
# Searches for new physics in collision events using a statistical technique for anomaly detection

### S. V. Chekanov⋆

HEP Division, Argonne National Laboratory, 9700 S. Cass Avenue, Lemont, IL 60439, USA.

⋆ chekanov@anl.gov

## Abstract

**This paper discusses a statistical anomaly-detection method for model-independent searches for new physics in collision events produced at the Large Hadron Collider (LHC). The method requires calculations of $Z$-scores for a large number of Lorenz-invariant variables to identify events that deviate from those expected for the Standard Model (SM).**

## 1 Introduction

Searches for new physics at the Large Hadron Collider (LHC) are typically performed using a model-dependent approach that involves Monte Carlo events created according to Standard Model (SM) predictions ("background") and "signal" events using hypothetical models beyond the Standard Model (BSM). In this technique, BSM Monte Carlo simulations guide the analyzers to identify kinematic regions which are affected by proposed BSM models. It should be pointed out that signatures of BSM physics may be sufficiently unusual, or more subtle than we usually anticipate and, therefore, may not be fully covered by model builders.

This paper discusses a simple model-agnostic technique that does not require complex simulations of Monte Carlo models, nor specific expectations for BSM physics. It hypothesizes that new physics may produce unexpected signatures (such as peaks in invariant masses) hidden in the large SM backgrounds. To find such BSM events, one can select uncharacteristic events ("outliers") and look at their signatures. It is assumed that the algorithms for the outlier detection must not bias the signatures themselves that are used to claim a new physics.

The BSM searches using anomaly detection can proceed via a few steps: (1) Define an input ("feature") space for events using data; (2) Apply an anomaly-detection algorithm to this feature space using statistical or machine learning (ML) methods; (3) Define anomalous events (outliers); (4) Study physics distributions in the outlier event sample ("unblind"). The last step can focus on distributions that do not require precise knowledge of SM backgrounds.

For example, one can look for evidence of contributions from resonant BSM phenomena. On the technical side, this would require an analysis of invariant masses in the outlier sample. New states with two-body decays may introduce localized excesses in such distribution, which can be found without using Monte Carlo simulations for background modeling.

Input data for anomaly-detection algorithms for particle colliders should reflect the fact that different types of particles (or more complex objects, such as jets, $b$-jets) can be copiously produced. On the computational side, the lists holding the information about such particles or jets have variable sizes, i.e. they change from event to event. One possibility to deal with the varying-size data is to "map" experimental data to the fixed-size data structures, such as the rapidity-mass matrix (RMM) [1,2] designed to represent a large number of Lorentz-invariant observables in the form of a fixed-size data structure. Due to the unambiguous mapping of a large number of experimental signatures to the specific RMM cells, this event transformation allows the usage of a broad range of machine-learning techniques. In this paper, however, we will focus on a simple statistical procedure for anomaly detection that can be directly applied to the RMM values.

The $Z$-score method is a simple and widely accepted method to estimate how many standard deviations away a given observation is from the mean value, i.e. $Z = (x - \bar{x})/\sigma$, where $\bar{x}$ is the mean of a variable $x$ and $\sigma$ is the standard deviation. In the case of the RMM input, one can build the following two measures for possible deviations:

- Outlier detection using the so-called Stouffer's $Z_S$-score method. First, $Z_{ij}$ is calculated for each RMM cell at the position $(i, j)$: $Z_{ij} = (X_{ij} - \overline{X}_{ij})/\sigma(X_{ij})$ where $X_{ij}$ is the value of the RMM cell, and $\overline{X}_{ij}$ and $\sigma(X_{ij})$ are the mean value and standard deviation calculated for all events. Since $X_{ij}$ values have low correlations [1] for SM events, the global $Z_S$ score for $N$ cells of the RMM can be calculated using the Stouffer's method of combining $Z$ scores: $Z_S = \sum_{i=1}^{N} Z_{ij}/\sqrt{N}$.

- Alternatively, one can sum up all the values of the RMM matrix, and then calculate the "event" $Z$ score: $Z = (X - \overline{X})/\sigma(X)$, where $X = \sum_{i=1}^{N} Z_{ij}$. This sum explicitly depends on the total number of active RMM cells.

We expect that $Z_S$ values are sensitive to deviations of RMMs from the mean values (expressed in the terms of standard deviations), but they are much less sensitive to the number of activated cells ($N$). In contrast, $Z$ should have a large sensitivity to events with a large multiplicity of jets or identified high-$p_T$ particles.

Anomaly detection using the $Z$-score method can be illustrated using Monte Carlo (MC) simulations. The MC event samples generated with Pythia8 [3] were taken from the previous studies [2]. We use the Charged Higgs process ($H^+ t$) as a representative BSM model with the signatures that are expected to be similar to SM events if they are selected with at least one lepton. The events were transformed to the RMMs with five types ($T = 5$) of the reconstructed objects: jets, $b$-jets, muons, electrons and photons. Up to ten particles per type were considered ($N = 10$), leading to the so-called T5N10 topology for the RMM inputs. Then, the two $Z$-scores defined above were calculated using the mean and $\sigma$ values from the combined SM+BSM event sample assuming the fractions $0.82 : 0.18$ for the dijet QCD and $W/Z/top$ events, respectively. These fractions were estimated using Pythia8 after requiring a single lepton. About 1M SM events were used. Then, 1000 $H^+ t$ events (0.1% of the SM event rate) were added to simulate events with a small contribution from BSM physics.

Figure 1 illustrates the shapes of distributions of the two $Z$-scores for the SM and $H^+ t$ processes. It shows that the outlier events with large (small) $Z$-scores have significant contributions from the charged Higgs boson process. We expect that this conclusion holds for a large number of BSM models.

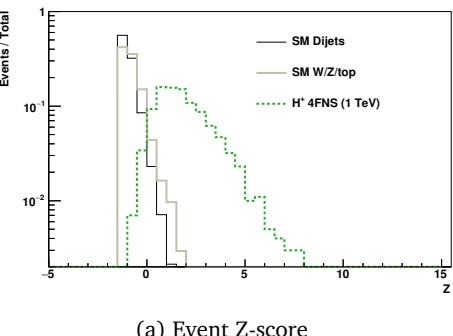
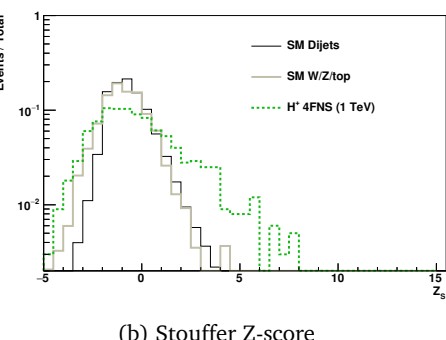

(a) Event Z-score          (b) Stouffer Z-score

Figure 1: Distributions of the Z-score values in Monte Carlo simulations for the SM processes (SM Dijets and $W/Z/top$) and for the charged Higgs boson ($H^+t$) process with the $H^+$ mass of 1 TeV. All events were pre-selected by requiring at least one lepton with $p_T^l > 60$ GeV.

Note that the $Z$-score calculations require two passes over data. The first pass is needed to calculate the averages used in the $Z$ definition. After the outlier events are selected using either small or large values of $Z$, one can study invariant masses in the outlier events, assuming that BSM events will likely populate the outlier, and can produce enhancements above smoothly falling backgrounds. For example, outlier events can be defined by $Z_S(Z) > 5$, or by some combinations of both $Z$ and $Z_S$. The invariant masses in question can be excluded from the RMM used for the $Z$-score evaluation to avoid biases that can influence the shape of smoothly falling invariant-mass distributions. Note that if the bump-hunting is performed in a large number of invariant masses included in the right-top corner of the RMM, then the criteria for discoveries should be significantly higher than for a standard observation of a bump in a single histogram. This is because of the fact that the look-elsewhere-effect contributing to the statistical significance of such observations can be non-negligible.

We expect that ML techniques (such as autoencoders) with RMM inputs for outlier detection will overperform the statistical $Z$-score method since ML can take into account correlations between RMM variables. They are ignored in the $Z$-score method. However, the statistical $Z$-score method would benefit from faster data processing.

**Acknowledgements.** I would like to thank W. Hopkins and T. LeCompte for the discussion. I gratefully acknowledge the computing resources provided by the Laboratory Computing Resource Center at Argonne National Laboratory. The submitted manuscript has been created by UChicago Argonne, LLC, Operator of Argonne National Laboratory ("Argonne"). Argonne National Laboratory's work was funded by the U.S. Department of Energy, Office of High Energy Physics under contract DE-AC02-06CH11357.

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
