# Peer review of "Searches for new physics in collision events using a statistical technique for anomaly detection"

_SciPost Physics Proceedings, doi:SciPost Phys. Proc. 10, 015 (2022)_

## Round 1 · Referee Report · Anonymous (Referee 1) · 2022-1-28

Strengths

A clearly written paper, giving specifics of the simple statistical method and an example of good performance.

Weaknesses

Appropriate for proceedings.

Report

While not a block for proceedings publication, I note:

0) The RMM is introduced in the author's previous papers, but not really explained here despite not being a widely known concept. But there was a page limit...

1) The effective assumption of Gaussian X_i distributions for RMM seems a bit dangerous, since non-Gaussian tails could masquerade as BSM outliers. I think this can be remedied by a more explicit calculation of full empirical p-value from the found distribution, and conversion to Z, though.

2) The Fig 1 performance plots seem equally normalised across samples with very different cross-sections... this isn't quite clear from the presentation, and a log plot with more realistic cross-sections would illustrate the physics power better, especially re. how the BSM distribution compares to the SM tails.

Requested changes

None needed.

---

## Editorial Decision

published